# Amino Acid Nanofibers Improve Glycemia and Confer Cognitive Therapeutic Efficacy to Bound Insulin

**DOI:** 10.3390/pharmaceutics14010081

**Published:** 2021-12-29

**Authors:** Aejin Lee, McKensie L. Mason, Tao Lin, Shashi Bhushan Kumar, Devan Kowdley, Jacob H. Leung, Danah Muhanna, Yuan Sun, Joana Ortega-Anaya, Lianbo Yu, Julie Fitzgerald, A. Courtney DeVries, Randy J. Nelson, Zachary M. Weil, Rafael Jiménez-Flores, Jon R. Parquette, Ouliana Ziouzenkova

**Affiliations:** 1Department of Human Sciences, The Ohio State University, Columbus, OH 43210, USA; lee.7278@osu.edu (A.L.); drshashikumar81@gmail.com (S.B.K.); kowdley.2@buckeyemail.osu.edu (D.K.); leung.167@buckeyemail.osu.edu (J.H.L.); muhanna.2@buckeyemail.osu.edu (D.M.); 2Department of Chemistry and Biochemistry, The Ohio State University, Columbus, OH 43210, USA; mason.870@osu.edu (M.L.M.); lin.1338@buckeyemail.osu.edu (T.L.); sun.596@buckeyemail.osu.edu (Y.S.); parquette.1@osu.edu (J.R.P.); 3Department of Food Science and Technology, The Ohio State University, Columbus, OH 43210, USA; ortegaanaya.1@osu.edu (J.O.-A.); jimenez-flores.1@osu.edu (R.J.-F.); 4Department of Biomedical Informatics, The Ohio State University, Columbus, OH 43210, USA; lianbo.yu@osumc.edu; 5Department of Neuroscience, The Ohio State University, Columbus, OH 43210, USA; Julie.Fitzgerald@osumc.edu (J.F.); Courtney.devries@hsc.wvu.edu (A.C.D.); zachary.weil@hsc.wvu.edu (Z.M.W.); 6Department of Neuroscience, West Virginia University, Morgantown, WV 26506, USA; 7Rockefeller Neuroscience Institute, West Virginia University, Morgantown, WV 26506, USA; randy.nelson@hsc.wvu.edu

**Keywords:** diabetes, leptin, nanofibers, taurine

## Abstract

Diabetes poses a high risk for debilitating complications in neural tissues, regulating glucose uptake through insulin-dependent and predominantly insulin-independent pathways. Supramolecular nanostructures provide a flexible strategy for combinatorial regulation of glycemia. Here, we compare the effects of free insulin to insulin bound to positively charged nanofibers comprised of self-assembling amino acid compounds (AACs) with an antioxidant-modified side chain moiety (AAC2) in both in vitro and in vivo models of type 1 diabetes. Free AAC2, free human insulin (hINS) and AAC2-bound-human insulin (AAC2-hINS) were tested in streptozotocin (STZ)-induced mouse model of type 1 diabetes. AAC2-hINS acted as a complex and exhibited different properties compared to free AAC2 or hINS. Mice treated with the AAC2-hINS complex were devoid of hypoglycemic episodes, had improved levels of insulin in circulation and in the brain, and increased expression of neurotransmitter taurine transporter, *Slc6a6*. Consequently, treatment with AAC2-hINS markedly advanced both physical and cognitive performance in mice with STZ-induced and genetic type 1 diabetes compared to treatments with free AAC2 or hINS. This study demonstrates that the flexible nanofiber AAC2 can serve as a therapeutic platform for the combinatorial treatment of diabetes and its complications.

## 1. Introduction

Insulin saves the lives of patients with insulin-dependent type 1 diabetes (T1D) and is the primary treatment for hyperglycemia in advanced stages of type 2 diabetes (T2D) [1]. Glycemic control has been significantly improved over the last decade through the application of continuous blood glucose monitoring, as well as the ongoing development of nanomaterials that release insulin in response to changes in blood glucose concentrations [2]. However, clinical outcomes of continuous glucose monitoring in patients with basal insulin treatment has not reduced the rate of hospitalization of patients with diabetes [3]. Neurodegeneration is a major complication that continues to contribute to hospitalization and mortality in patients with diabetes, which remains as high as cancer deaths [4].

Current strategies to improve control over insulin action follow three major directions: (1) identifying insulinotropic hormones, (2) modifying the structure of insulin to improve its tissue specific responses and efficacy, and (3) creating polymers and microdevices capable of releasing insulin in response to elevated blood glucose [1]. However, various non-glycemic adverse effects (reviewed in [5]), including insulin-induced vasodilation, changes in vascular permeability, and sodium retention contribute to the low therapeutic index of insulin, calculated as a ratio of beneficial to adverse effects [1]. One of the major complications of diabetes is the damage of nervous tissue. Insulin impedes the cleavage of beta amyloid by the insulin-degrading enzyme in the brain [6], resulting in the formation of amyloid aggregates [7], and an increased tau phosphorylation [8]. Normoinsulinemia supports cognition, whereas both hypoinsulinemia and, particularly, supraphysiological concentrations of insulin impair cognition and worsen dementia pathology [9,10,11]. Insufficient control over the action of therapeutic insulin is now considered as a major reason behind its adverse effects (reviewed in [1]).

The brain utilizes 25% of the total metabolic glucose for energy [12]. The fundamental difference between the glucose supply in the nervous and peripheral tissues is the utilization of over nine different glucose transporters [13] compared to the major insulin-regulated GLUT4 in peripheral adipose and muscle tissues. Among them, glucose transporter 1 (GLUT1) is the major transporter for glucose in the brain [13,14]. Nevertheless, in the brain, GLUT4 activation regulates glucose sensing and glucose tolerance [15] in conjunction with other glucose transporters in the brain. Chronic diabetic conditions progressively diminish glucose transport, which leads to impaired brain growth and cognitive development in children with T1D [16,17]. Adult patients with T1D, who experience hyper- and hypoglycemic episodes, have a six-fold greater risk for the development of dementia compared to those without these complications [18]. The high prevalence of neurocognitive dysfunction and insulin resistance in the brain is also associated with diabetes mellitus, which is often described as ‘type 3 diabetes’ [19]. The management of the neurological and cognitive complications of T1D [1] and T2D [20] represents a continuing challenge, which may be a consequence of insulinocentric therapies for the treatment of diabetes.

A century of research supports a combinatorial model of insulin signaling whereby a single hormone produces a range of responses [5,21]. These diversified effects not only depend on crosstalk within downstream intracellular signaling pathways [22], but also involve changes in membrane polarization [23], the composition of lipid mediators [22], and the release of other hormones, such as leptin [24] for regulation of GLUT1 and/or other transporters in brain [25] and other tissues [26]. The integration of insulin signaling with leptin and other pathways is necessary and required for efficient glucose uptake [27,28]. Scaffold proteins commonly play a critical role in the combinatorial regulation by localizing and assembling components of a signaling pathway in specific parts of the cell in a protective milieu (reviewed in [29,30]). To address the specific glycemic requirements of the nervous system under insulin-deficient and resistant states, we have previously developed and tested amino acid compound 2 (AAC2) [31]. Glucose utilization was activated by binding of AAC2 to alternative sites on the leptin receptor (LepR) and was sufficient to treat mice with T1D and T2D, and to reduce anxiety in diabetic mice [31]. In addition, the molecule, AAC2, self-assembled into nanofibers with a positively charged surface [31]. However, the ability to bind negatively charged insulin and serve as its scaffold has not been explored. In this proof-of-concept study, we hypothesized that the AAC2 nanofibers with surface-bound insulin could induce combinatorial effects on the regulation of glucose metabolism in peripheral and nervous tissues and mitigate cognitive decline in mice with insulin-deficient diabetes.

## 2. Materials and Methods

### 2.1. Materials

We purchased from Sigma-Aldrich (St. Louis, MO, USA) phosphate-buffered saline (PBS, D8537), recombinant human insulin solution (hINS, I9278), streptozotocin (STZ, S0130). GLUT1 inhibitor (BAY-876, Selleckchem, Houston, TX S8452, USA)

### 2.2. Synthesis of AAC Compounds

The syntheses and characterization of AAC2 or AAC6 compounds were described in [31]. AACs were sterilized by X-ray irradiation at 748 cGy/min for 30 min by an RS 2000 irradiator (Rad Source Technologies; Suwanee, GA, USA). An amount of 0.1 mM AAC2 in PBS was stored at 4 °C until it was used.

Preparation of AAC2 bound to insulin: 100 µL of 0.1 mM AAC2 solution was mixed with 100 µL hINS (10 mg/mL) into 800 µL of PBS (total 1 mL). This combination (AAC2-hINS) was stabilized for 30 min at RT before injection.

### 2.3. Transmission Electron Microscopy (TEM)

AAC2 or AAC6 was dissolved in PBS to form a 20 mM solution and left for 12 h. After gelation, the solution was diluted to 1 mM, and 30 µL of the diluted samples were loaded on the formvar/carbon-covered copper grid (Ted Pella, 01801, Redding, CA, USA) and stained with uranyl acetate (1 wt% in distilled water) for 30 s and the grids were dried with filter paper. The structures of the AAC2 or 6 were observed by TEM (FEI Tecnai G2 Spirit, Thermo Fisher, Waltham, MA, USA).

### 2.4. Zeta Potential Measurement

Zeta potentials were measured using the Malvern Zetasizer Nano ZS system. Titration with hINS was conducted with 0.1 µM of AAC2 or AAC6.

### 2.5. Quartz Crystal Microbalance with Dissipation (QCM-D) Binding Assay

A quartz sensor with an active gold surface (QSX 301, Biolin Scientific, Beijing, China) and a fundamental frequency of 4.95 MHz was first equilibrated with PBS buffer pH 7.1–7.5 in a Q-Sense Explorer instrument (Biolin Scientific) at a flow rate of 50 µL/min and 25 °C until a stable baseline was observed. Afterwards, a diluted solution of AAC2 (0.1 μM in PBS) was passed through the sensor at the same flow rate and temperature. The molecular absorption was monitored using QSoft 401 software (Biolin Scientific) until no changes in ΔF and ΔD were observed indicating stabilization of the mass adsorbed. Subsequently, hINS (10 μg/mL PBS) was passed through the sensor. The interaction of hINS with AAC2 was studied under the same conditions. The interaction of each component layer, measured as a difference in frequency and dissipation values of the odd overtones was modeled using Voigt–Voinova equations for homogenous viscoelastic layers [32,33] assuming a fixed density of 1 g/mL using QTools 3 software (Biolin Scientific). The values of thickness in real time were calculated to corroborate molecular interaction between AAC2 and hINS.

### 2.6. Spectroscopy

UV-vis experiments were performed on a Shimadzu UV-2450 Spectrophotometer in a 1 mm or 3 mm quartz cuvette. Fluorescence emission experiments were performed on a Shimadzu RF-5301PC Spectrofluorophotometer in a triangular or 3 mm square quartz cuvette. Excitation and emission slit widths were set to 1.5 mm in all experiments.

### 2.7. Confocal Fluorescence Microscopy

Samples in PBS were deposited onto 1.0 mm glass microscope slides with #1.5 glass cover slips, sealed on the edges with clear liquid nail polish. Imaging was performed using a Nikon A1R Live Cell Confocal microscope equipped with a 32-channel PMT spectral detector. Samples were excited with either 402 nm or 487 nm laser lines, as indicated per experiment, with emission wavelengths collected in spectral detector mode with separation into 32 channels with a resolution of 6 nm. Images were collected with a 100× oil immersion objective and processed using NIS-Elements analysis software and FIJI imaging software.

FRET Efficiency Calculations: Energy transfer efficiencies were calculated using Equation (1):(1)$$E=1-\frac{F_{\mathrm{DA}}}{F_{D}}$$
where E is the energy transfer efficiency, F_D_ is the fluorescence intensity of the donor, and F_DA_ is the fluorescence intensity of the donor in the presence of the acceptor at the same concentration. The reported FRET efficiencies were generated from fluorescence intensities at 479 nm.

### 2.8. Glucose Uptake Assay

Glucose uptake assay (Cayman Chemical, 600470, Ann Arbor, MI, USA) was performed in vitro using the published method [31]. Briefly, mouse 3T3-L1 fibroblasts/preadipocytes (CL-173, ATCC) were maintained in high glucose DMEM (Gibco, 11965) containing 10% new-born calf serum (Gibco, 26010) and 1% Penicillin-Streptomycin (10,000 U/mL; Gibco, 15140). The 3T3-L1 cells (4 × 104/0.1 mL/well) were grown for 24 h in a 96-well plate. Starvation conditions were induced in cells washed with PBS and cultured in glucose-, L-glutamine-, and phenol red-free DMEM medium (Gibco, A14430, 200 µL/well) for 40 min. Fluorescent glucose 2-deoxy-2-[(7-nitro-2,1,3-benzoxadiazol-4-yl)amino]-D-glucose) (2-NBDG or FD glucose; Cayman Chemical, Ann Arbor, MI, 600470) was prepared in the glucose-free medium (FD-working solution, 0.29 mM). Cells were treated with reagents in the FD-working solution at 37 °C for 90 min. Fluorescence was measured in cells washed twice with PBS in 100 µL of PBS/well at an excitation/emission wavelength of 485/535 nm using Synergy H1 Hybrid Multi-Mode Microplate Reader (BioTek, Winooski, VT, USA).

Human brain endothelial cells (hBEC, CRL-3245, ATCC) were cultured in DMEM:F12 (30-2006, ATCC) supplemented with 10% FBS (10082, Gibco, Gaithersburg, MD) and endothelial cell growth supplement (component of CCM027, R&D systems, Minneapolis, MN). The hBEC (2 × 104/0.1 mL/well) were cultured in a 96-well plate coated with 0.1% gelatin solution (Sigma-Aldrich, ES-006-B) for 24 h prior to glucose uptake measurement. Cells washed with PBS, were treated with vehicle (Veh; PBS) or hINS (1.7 µM) or AAC2 (0.1 µM) or AAC2-hINS which were diluted in same glucose free DMEM (Gibco, A14430, 200 µL/well) for 40 min. Then, GLUT1 inhibitor (BAY-876; 10 nM diluted in DMSO; Selleckchem, Houston, TX, S8452) or DMSO were added in 100 µL of FD-working solution per well and incubated for 50 min at 37 °C.

### 2.9. Transfected Cells

Non-differentiated NIH-3T3 cells were transiently transfected with GLUT4-GFP plasmid (Addgene #52872) using FuGene 6 transfection reagent (E2311, Promega, Madison, WI, USA) as previously described [34]. Transfected cells were plated on Petri-35 dishes with coverslip (Cat #P35G-1.5-14-C, MatTek, Ashland, MA, USA). GLUT4-GFP translocation was demonstrated using confocal microscopy (Olympus FV10i).

### 2.10. Animal Studies

Animal studies were approved by the Institutional Animal Care and Use Committee of The Ohio State University (OSU). All mice were purchased from The Jackson Laboratory (Bar Harbor, Maine). Mice were fed a regular chow diet (Teklad LM-485 mouse/rat diet, irradiated; Envigo, Somerset, NJ, USA) under 12 h:12 h light:dark cycle. Mice were sacrificed by isoflurane inhalation followed by cardiac puncture.

#### 2.10.1. Experimental Mouse Models

Chemically induced T1D mouse model. Five-week-old male C57BL/6J mice were used (wild type, WT, JAX. Stock number 000664). After a week of the assimilation period, T1D was induced in fasted mice by five daily intraperitoneal injections (i.p.) of STZ (50 mg/kg BW, pH 4.5) [35,36]. Diabetes was confirmed by glucose levels of 13.8 mM at 2 days after the last STZ injection [35], Compared to the levels of fasting glucose in healthy mice of the same age 9.0 ± 2.1 mM (*n* = 6). Three days after the last STZ injection, mice were assigned to four treatment groups. Treatments were: (1) control (10 µL PBS/g BW), (2) AAC2 (10 µL 0.1 nmol AAC2/g BW), (3) hINS (10 µL 10 µg = 1.7 nmol hINS /g BW, and (4) hINS bound to AAC2 (AAC2-hINS, 10 µL 0.1 nmol AAC2/1.7 nmol hINS/g BW or 1:17 AAC2: hINS mol/mol). I.p. injections of the treatments were administered every three days to non-fasted mice. Three independent studies were performed with mice from different litters. Age-matched wild type male mice (WT; C57BL/6J. JAX stock number 000664) were included for comparison.

Genetically modified T1D mouse model. The 5- or 6-week-old male C57BL/6-Ins2Akita/J, (JAX. Stock number 003548) were intraperitoneally injected (i.p.) every 48 h with 10 µL PBS/g BW or therapeutic reagents (10 µL AAC2 0.2 nmol/g BW and/or hINS 1.7 nmol /g BW). AAC2-hINS was prepared by mixing 200 µL of 0.1 mM AAC2 solution with 100 µL of hINS (10 mg/mL = 1.7 µmol/mL) into 700 µL of PBS. After 30 min incubation, the solution was injected 10 µL per gram body weight (concentrations 0.0.2 nmol of AAC2 bound with 1.7 nmol hINS/g BW; 1:8.5 mol/mol ratio).

#### 2.10.2. Body Composition and Activity Measurements

Mice body composition was measured with an EchoMRI™-100H Body Composition Analyzer for Live Small Animals (EchoMRI™, Houston, TX). A Comprehensive Lab Animal Monitoring System (CLAMS) was used. Metabolic parameters including XYZ movement activity were measured by indirect calorimetry (CLAMS, Columbus Instruments, Columbus, OH, USA) at an ambient temperature (22 °C) with 12 h light/dark cycles 7 weeks after treatment.

#### 2.10.3. Glucose Tolerance Test (GTT) and Insulin Tolerance Tests (ITT)

After 3 weeks from the beginning of treatments, GTT was performed to 5 h-fasted mice by i.p. injection of 10% glucose solution (10 µL/g BW). For ITT, fasted mice were injected with a single i.p. insulin dose (1 mU of hINS/g BW). Mice were allowed to recover for one week between GTT and ITT tests.

#### 2.10.4. Cognitive Test

Y-maze test. Spontaneous alternation is the tendency to choose other options than one previously chosen. Mice were placed in the center of the Y-maze for 2 h after lights had been off for 8 min. The number of alternating 3-choice sequences over the total chances for each mouse had to alternate between arms was recorded to analyze the alternation percentage (% SA) [37].

Barnes maze test. The Barnes maze test was performed as described in [38] at the Behavioral Core facility at OSU. The Barnes maze, (122 cm diameter) with 18 escape holes (9.5 cm) placed every 20° around the perimeter (ENV-563-R, MedAssociates, St. Albans, VT, USA), was surrounded with a 60 cm high white polycarbonate barrier to prevent escape. Black panels blocked the blind escape holes and the target escape hole was visually the same as the blind holes but contained a black escape box (38.7 × 12.1 × 14.2 cm). Distinct visual cues (black 2-dimensional geometric shapes, 20–25 cm) on the upper edge were attached to the surround at the 4 compass points and present visual cues distal to the maze. Testing consisted of 5 days of acquisition training followed by a single probe trial 24 h after the last training trial. Each acquisition day consisted of one session/animal, 3 trials per session, with an inter-trial interval of 5 min. For acquisition training, all mice were allowed to acclimate for 30 min before the start of testing. Each trial consisted of carefully placing the mouse in the center of the maze from the opaque plastic beaker. The mouse was allowed to search for the escape box for 120 s, then it was guided to it. Olfactory cues were eliminated by cleaning with 70% ethanol after testing each mouse, and each day the maze was rotated 90° counter-clockwise, with the escape box location and location of visual cues remaining constant throughout testing. All behavior on coded mice were recorded and scored using The Observer software (XT 8.0; Noldus, Leesburg, VA, USA). For training trials, latency to escape and number of errors were recorded. An error was defined as an investigation of a blind escape hole where the entire head of the mouse broke the plane of the edge of the escape hole. For the probe trial, latency to escape hole, number of errors, and time in quadrant of escape hole (% in path Q3) were measured.

### 2.11. Biochemical Analysis

Aspartate aminotransferase (AST) activity was measured in mice plasma using an assay from Abcam (ab105135, Cambridge, UK). Mouse insulin was measured in plasma by enzyme-linked immunosorbent assays (ELISA; EMD Milipore, EZRMI-13K, Billerica, MA). The level of human insulin in mice plasma was analyzed using ELISA (EMD Millipore, EZHI-14K). Plasma mouse leptin was measured by ELISA (Crystal Chem, 90030, Downers Grove, IL, USA). Hemoglobin A1c (HbA1c) level of whole blood was analyzed by an enzymatic assay according to manufacturer’s instruction (Crystal Chem, 80310, Downers Grove, IL, USA). All ELISAs were performed according to the manufacturer’s instructions. The absorbance from each biochemical analysis was measured using a Synergy H1 Hybrid Multi-Mode Microplate Reader (BioTek, Winooski, VT, USA).

### 2.12. mRNA Analysis

Total mRNA was purified from mouse brain according to the manufacturer’s instructions (Qiagen, Germantown, MD, USA). RNA integrity was analyzed using the Agilent 2100 Bioanalyzer (Agilent Technologies, Santa Clara, CA, USA). mRNA was quantified using the 12 K Flex QuantStudio Real-Time PCR System and TaqMan fluorogenic detection system (Applied Biosystems/Thermo Fisher). To examine the expression of genes, we used validated mouse Ins1 (Mm01950294_s1) and Ins2 (Mm00731595_gH) TaqMan probes, which were also purchased from Thermo Fisher. Comparative real-time PCR was performed in triplicate, including no-template controls. Expression was calculated using the comparative Ct method using mouse housekeeping gene hypoxanthine-guanine phosphoribosyl transferase (*Hprt*, Mm03024075_m1).

### 2.13. Affymetrix GeneChip

RNA was isolated by Rneasy kit (Qiagen, 74106, Hilden, Germany). To purify RNA, Rnase-Free Dnase Set was used (Qiagen, 79254). RNA integrity was measured using the Agilent 2100 BioAnalyzer (Agilent Technologies, Santa Clara, CA, USA). A 100 ng aliquot of total RNA was linearly amplified. Then, 5.5 µg of cDNA was labeled and fragmented using the GeneChip WT PLUS reagent kit (Affymetrix, 902280, Santa Clara, CA, USA) following the manufacturer’s instructions. Labeled cDNA targets were hybridized to Affymetrix GeneChip Mouse Gene ST 2.0 arrays for 16 h at 45 °C rotating at 60 rpm. The arrays were washed and stained using the Fluidics Station 450 (Thermo Fisher, 00-0079, Waltham, MA) and scanned using a GeneChip Scanner 3000 (Thermo Fisher, 00-0210). Signal intensities were quantified by Affymetrix Expression Console version 1.3.1 (Thermo Fisher). Background correction and quantile normalization were performed to adjust for technical bias, and probe-set expression levels were calculated by the RMA method. After filtering above noise cutoff (*p* < 0.01), there were 9528 probe-sets tested by linear model. A variance smoothing method with fully moderated t-statistic was employed for this study and was adjusted by controlling the mean number of false positives. QIAGEN Ingenuity Pathway Analysis (QIAGEN IPA) was used to identify pathways that were significantly affected by treatments.

### 2.14. Statistical Analysis

All data were analyzed using SPSS 23 (IBM Corp. in Armonk, NY, USA). All data are shown as mean ± SEM. A number of samples for each assay is indicated in Figure legends. Group comparisons were assessed using an unpaired *t*-test (two-sided) or one-way analysis of variance (ANOVA) for normally distributed samples. Mann–Whitney U-test or Kruskal–Wallis test were used as nonparametric tests. *p* < 0.05 was considered statistically significant.

## 3. Results and Discussion

### 3.1. AAC2 and Insulin Form a Supramolecular Complex In Vitro

In a previous study, we described the properties of AAC2 and demonstrated AAC2-dependent mechanisms that increase glucose uptake under insulin-resistant and insulin-deficient conditions in vivo [31]. This work exploited the cornerstone property of AAC2 to self-assemble into uniform, nanometer-scale, positively charged nanofibers [31]. AAC2 nanofibers have a positive Zeta potential at biological pH (Figure 1a,b). To demonstrate the AAC2 interaction with negatively charged insulin (isoelectric point at pH 5.3, [39]) we measured Zeta potential as a function of the ratio of hINS to AAC2 (Figure 1b). The initially positive Zeta potential of AAC2 progressively decreased as the ratio increased with hINS addition, suggesting a strong electrostatic interaction between AAC2 and hINS in vitro. As a control, we evaluated the interaction between AAC6, which does not assemble in PBS, and hINS by recording the Zeta potential as a function of the AAC6/hINS ratio, which indicated a poor interaction between the two molecules (Figure 1c). AAC6 was comprised of positively charged lysine residues but lacked the ability to form nanofibers. In this experiment, the initially negative Zeta potential of hINS was not affected by AAC6 (Figure 1d). This experiment suggested that the formation of positively charged nanofibers was necessary to bind insulin.

Next, we analyzed whether the electrostatic binding of hINS to the surface of the AAC2 nanofibers produced a stable complex. The kinetics of the formation of AAC2 complex with hINS was assessed using quartz crystal microbalance experiments. In response to a generated acoustic wave, changes in the frequency (ΔF) are directly proportional to the added mass and thickness while changes in the dissipation energy (ΔD) contain information about the viscoelastic properties of the film [40]. The changes in frequency and dissipation in response to.AAC2 indicated the formation of a stable film (Figure 1e), which bound insulin to form a 15 nm film on the active gold surface at physiological conditions (Figure 1f). This film was stable after a PBS wash, suggesting that insulin was not released from the nanofibers (Figure 1e,f) and thus formed a stable complex.

The binding of hINS with AAC2 was further investigated using fluorescence resonance energy transfer (FRET). Due to their significant spectral overlap, the coumarin and fluorescein chromophores have been used as pairs that undergo FRET when positioned proximal to each other [41,42,43]. The diethylaminocoumarin (DAC) chromophore of AAC2 and fluorescein isothiocyanate (FITC)-labeled insulin (FITC-INS) exhibited significant spectral overlap (Figure 2a), which was sufficient to investigate the interactions between AAC2 and FITC-labeled insulin using FRET spectroscopic experiments and confocal fluorescence imaging. Accordingly, FRET experiments were performed at different molar ratios of FITC-INS to a constant concentration of AAC2 (Figure 2b). Figure 2b shows a decrease in the AAC2 fluorescence at 479 nm with increase in FITC-INS concentrations. The calculated FRET efficiencies of each mixture revealed a significant logarithmic relationship between energy transfer and FITC-INS concentrations (Figure 2c). The efficient energy transfer from the AAC2 donor to the FITC acceptor indicated that AAC2 and FITC-INS were in close proximity, due to complex formation. When one mixture with a fixed ratio (1:0.67 AAC2:FITC-INS) was diluted with PBS to concentrations as low as 390 nM, the FRET efficiency was still present, but much lower (Figure 2d). This observation revealed a critical micelle concentration that was significantly lower than the value (84 μM) measured for free AAC2 nanofibers [31]. The observation of FRET between AAC2 and FITC at concentrations significantly lower than 84 μM suggested that the AAC2-INS complex was more stable at lower concentrations than AAC2 nanofibers alone.

AAC2 (1 mM), FITC-INS, and a 10:1 mixture of AAC2:FITC-INS were imaged by confocal fluorescence (Figure 2e–i). Images of a 1 mM sample of the AAC2 nanofibers exhibited strong emission at 487 nm when excited at 402 nm (Figure 2f,g). The emission profile generated from the image overlapped well with the fluorescence spectra of AAC2. In contrast, no aggregated species were detected in samples of FITC-labeled insulin alone (Figure 2e). Strikingly, images of AAC2 with bound FITC-INS revealed strongly emissive nanofibers when excited at both 402 and 487 nm (Figure 2h). The emission profile from AAC2 with bound FITC-INS excited at 402 nm (coumarin excitation) showed two peaks, a small peak at 465 nm and a larger peak at 537 nm, while FITC excitation at 487 nm showed an emission peak only at 537 nm (Figure 2i). Co-plotting the emission collected from the confocal images with the spectra of AAC2 and FITC-INS (Figure 2i), indicated that the emission of the assembled structures in the mixture came predominantly from the FITC, with a smaller contribution of AAC2 fluorescence detectable when excited at 402 nm due FRET energy transfer from AAC2 to FITC-INS. The quenching of AAC2 fluorescence via FRET between AAC2 and FITC-insulin and the observation of FITC emission from the AAC2 nanofibers indicated that the pre-assembled AAC2 structures bind FITC-INS. Together, the formation of the AAC2-hINS complex was demonstrated by changes in Zeta potential, microbalance kinetics, and FRET interactions with the AAC2-hINS ratio.

### 3.2. AAC2-hINS Complex Stimulated Glucose Uptake In Vitro in Peripheral and Nervous Tissues

To test how the formation of the AAC2-hINS complex altered the glycemic properties of AAC2 and hINS, we measured the uptake of FD-glucose by 3T3-L1 mouse fibroblasts/preadipocytes. Compared to non-treated cells (control, 100%), glucose uptake was significantly improved by treatment with free hINS (123%), as well as by free AAC2 at low (30 nM, 109%) and high concentrations (3 μM, 143%) (Figure 3a). Using a constant hINS concentration (10 mg/mL), in conjunction with AAC2, at 30 nM and 3 mM, led to an increase in glucose uptake to 137% and 129%, respectively. These data suggest that FD-glucose uptake reached a plateau at higher concentrations of AAC2 with or without hINS. Importantly, that AAC2-hINS complex achieves this maximal effect at a low nM concentration of AAC2. AAC2 with hINS significantly increased the uptake of FD-glucose at nanomolar concentrations of AAC2 (Figure 3a). Free AAC2 exhibited a dose-dependent uptake of glucose (Figure 3b). To assess whether the binding of hINS with AAC2 altered the mechanisms of glucose uptake, we transiently transfected NIH-3T3 cells with a vector overexpressing the major insulin-responsive glucose transporter 4 (GLUT4), which was labeled with green fluorescence protein (GLUT4-GFP) for imaging. These preadipocytes mimic peripheral tissues’ response to insulin. In response to free hINS, but not to free AAC2 stimulation, GLUT4-GFP was translocated to the membrane and became visible in fluorescent microscope (Figure 3c). The AAC2-hINS complex retained the ability to induce GLUT4 translocation in these cells. We also analyzed the effect of free AAC2, hINS and AAC2-hINS in brain endothelial cells (bEC), which utilize GLUT1 transporter, which is not dependent on insulin [44]. These cells comprise the brain–blood barrier (BBB). Accordingly, in human bEC, FD glucose uptake was mediated by AAC2 (Figure 3d), but not by free hINS in contrast to 3T3-L1 and NIH-3T3 cells (Figure 3a–c). Remarkably, binding of AAC2 with hINS amplified glucose uptake compared to either free AAC2 or hINS in human bEC. The increase in glucose uptake was abolished by the GLUT1 inhibitor BAY-878 (Figure 3d). These data suggest that binding between AAC2 and hINS synergistically improved glucose uptake by mechanisms involving GLUT4 and GLUT1 to support glucose uptake by both, neurovascular BBB and peripheral tissues.

### 3.3. Free AAC2, Free hINS, and AAC2-hINS Rescued STZ-Treated Mice from Death

Next, we investigated the effects of free and bound AAC2 and hINS in vivo under insulin-deficient conditions induced by streptozotocin (STZ)-mediated death of the pancreatic beta cells [36]. We used hINS for treatments to distinguish between endogenous and therapeutic insulin. We injected mice every 72 h with free hINS, AAC2, or hINS bound to AAC2 prior to the treatment (AAC2-hINS) to assess the duration of therapeutic action for each formulation. Given that the improved glucose uptake efficacy in 3T3-L1 cells has been observed at nanomolar concentrations of AAC2 (30 nM) and micromolar concentrations of insulin (1.7 μM) (Figure 3a), we used 1:17 AAC2: hINS mol/mol ratio in AAC2-hINS and used the same dose for free AAC2 and hINS in this study. The design and timeline of the study are outlined in Figure 4a. Three similar studies were performed with different litters of animals.

The onset of hyperglycemia characteristics for T1D [45] in young wild type mice (WT) after STZ injections was indicated by high levels of fasting glucose (Figure 4b). Afterwards, mice were randomized into treatment groups. Mice treated with free hINS, AAC2, or AAC2-hINS survived, grew, and gained weight for at least 100 days (Figure 4c, Appendix A). Mice treated with free AAC2 gained significantly less weight compared to the treatment groups receiving free and bound hINS; however, they remained functional throughout the 100-day intervention (the duration of study was limited by available funding, not by mice conditions). All STZ-treated mice in the control group developed severe diabetes and reached the elimination criteria (>20% weight loss, pain and distress) 30 days after beginning the first study; therefore, the duration of subsequent studies was confined to 30 days (Appendix A). Food intake was not significantly different among groups (Figure 4d). The increased survival rate was not associated with toxicity by treatments; blood levels of aspartate transaminase (AST) were similar among the groups (Figure 4e) and resembled mean levels seen in the healthy controls from the same litters of mice. In line with mouse growth characteristics, the total fat mass measured by Echo MRI was significantly higher in the AAC2-hINS-treated groups compared to control and free AAC2 treated mice, whereas changes in the free hINS-treated mice were not significant (Figure 4f). The lean mass was similar across all groups of STZ-treated mice (Figure 4g). These data indicated that treatment with the AAC2-hINS complex exhibited some differences compared to treatments with free AAC2, because mice gained weight in AAC2-hINS compared to AAC2-treated animals. The AAC2-hINS-treated mice maintained body fat at significantly higher levels than the non-treated group, and more more effectively than those treated with free hINS. These physiological properties of AAC2-hINS were largely similar to properties of free AAC2 or free hINS, with the exception of the distinct synergistic effect of AAC2-hINS on the body fat composition compared to either of its free constituents.

### 3.4. AAC2-hINS to Increases the Blood Levels of Insulin and Glucose Uptake in STZ Mice

We hypothesized that binding of hINS to the AAC2 nanofibers could reduce the rate of hINS clearance in the circulation compared to free hINS. Treatments were performed every 72 h to provide sufficient time for clearance of the injected hINS. The levels of hINS were 2.71 times higher in AAC2-hINS treated vs. free hINS treated animals (Figure 5a), even though they were measured 48 h after the last injection of the 30-day study. This difference became more pronounced (5.6 fold) after 100 days of treatment with AAC2-hINS 0.498 vs. 0.092 mU/mL in free hINS-treated animals, indicating the protective effects of this nanofiber scaffold. The control and AAC2-treated animals maintained similarly low levels of endogenous mouse insulin (Figure 5b).

High insulin levels in circulation pose the risk of hypoglycemia [1]. We performed an insulin tolerance test (ITT) to elucidate this effect in all groups of mice. Despite the differences in the levels of circulating hINS, mice in all treatment groups presented similar insulin sensitivity during ITT (Figure 5c). Glucose uptake was compared among treatment groups using a glucose tolerance test (GTT) (Figure 5d). The control STZ mice exhibited severe glucose intolerance (Figure 5d). Treatments with free AAC2 markedly improved glucose tolerance even though the levels of endogenous mouse insulin were similar in the control and AAC2-treated mice (Figure 5b). These glycemic effects of AAC2 are in agreement with previously published observations in other mouse models of T1D and T2D [31]. Notably, AAC2-hINS-treated mice exhibited the classic glucose uptake kinetics, in contrast to the incomplete glucose clearance in mice treated with free hINS (Figure 5d). Only AAC2-hINS treatment resulted in significantly lower levels of HbA1C levels in blood compared to control STZ group 30 days post-treatement expression (Appendix A), although HbA1C is a marker of long-term changes in the blood glucose concentrations. These data demonstrated that circulating AAC2-hINS has a unique ability to stimulate glucose uptake without the hypoglycemic episodes associated with high blood concentrations of free therapeutic insulin in previous reports [1]. All indices of improved glucose metabolism, including the glucose tolerance test, demonstrated the improved efficacy of AAC2-bound insulin compared to free hINS.

It is possible that the concentrations used in our studies were suboptimal and the AAC2-hINS complex could be stabilized at a higher molar ratio of AAC2, based on the Zeta potential and FRET studies, and could be improved in the future. The dose of AAC2:hINS (1:17, mol/mol) used in this study could lead to dissociation of the complex. However, in our study, the presence of scaffold was strongly supported by the markedly different stabilities of the bound and free forms of hINS in the circulation. The AAC2-hINS formulation circulated in blood at 2.7 higher concentrations than free hINS, 48 h after the injection. The presence of the AAC2-hINS complex partially accounted for these differences in concentrations over time as well as the absence of hypoglycemic episodes, a known side effect of free insulin overdose [1]. Establishing the stability and the tissue distribution of the native AAC2-hINS complex under the low therapeutic concentrations used in this study remains technically challenging. Currently, some predicted complexes were resolved in artificial ‘native’-like environments using cryogenic electron microscopy in a specific milieu [46], whereas isolation and quantification of these native complexes from tissues has not been successful. In our subsequent experiments, we compered gene expression to elucidate whether the AAC2:hINS complex-regulated pathways are similar to those of free AAC2 and hINS constituents or complex acted as a distinct therapeutic modality.

### 3.5. Insulin Eexpression in the Brain

Recent studies demonstrated that the expression of *Ins1* and *Ins2* genes in the brain [47] that could potentially contribute to insulin levels in this tissue and differences in systemic metabolic effects and cognitive responses in STZ mice treated with hINS, AAC2, and AAC2-hINS. In our study, the lowest expression of mouse *Ins1* gene in the brain was associated with free hINS treatment. *Ins1* expression was significantly higher in groups treated with AAC2 as compared to hINS. The levels in AAC2-hINS group were comparable to those treated with the free AAC2 and the control (Figure 5e). In contrast, the expression of *Ins2* gene was reduced in mice treated with free AAC2 compared to free hINS; however, treatment with AAC2-hINS increased *Ins2* expression relative to both free hINS and free AAC2-treated mice (Figure 5f). These studies demonstrated a unique effect of the AAC2-hINS complex on *Ins1* and *Ins2* expression in the brain, which resembled levels observed in healthy mice, unlike of free AAC2 of hINS.

### 3.6. Distinct Regulatory Effects of AAC2-hINS Compared to Free AAC2 and hINS

Given the role of the liver as a metabolic hub, we assessed the differences in the underlying mechanisms between free and bound AAC2 and hINS, by analyzing hepatic gene expression using an Affymetrix microarray (Figure 6). Hepatic tissues were randomly selected from each of our three independent studies. In response to the treatments, a significant change in gene expression occurred in a total of 25 genes in the AAC2-treated, 45 genes in hINS-treated, and 69 genes in AAC2-hINS-treated groups compared to the control STZ mice (Figure 6a). In agreement with the previously proposed mechanistic differences underlying AAC2 and hINS action [31], only three genes were identically influenced by both AAC2 and hINS treatments (Figure 6a). The most profound effect on gene regulation was observed in AAC2-hINS-treated mice, leading to significant differences in the expression of 60 genes, which were not affected by treatments with free AAC2 or hINS. Overlap in the regulation of expression of two and seven genes was seen in mice treated with AAC2-hINS, and free AAC2 and hINS, respectively. This gene regulation pattern suggested that the mechanism underlying AAC2-hINS action was distinct from those of free AAC2 or hINS.

We employed a pathway analysis of the gene expression data to obtain an unbiased identification of the key metabolic networks distinguishing AAC2-hINS function from each of its free components: AAC2 and hINS. The pathway analysis revealed that the key differences in regulation between AAC2 and AAC2-hINS treated mice involved the innate immunity protein [48] C-type, C lectin domain family 2 member D (*Clec2d*) and angiopoietin-like protein 8 (*Angpl8* or betatrophin [49]) gene expression (Appendix A). These genes were significantly reduced by free AAC2 but were markedly upregulated by AAC2-hINS. Although regulation of *Angpl8* by insulin has been reported [50], free hINS had no impact on either *Clec2d* or *Angpl8* in our study. In contrast, Ca^2+^-dependent actin-binding plastin 1 (*Pls1*) [51] and amyloidogenic leukocyte-cell-derived chemotaxin 2 [52] (*Lect2*) were upregulated by free AAC2 and suppressed by the AAC2-hINS complex. Free hINS did not influence these genes. Expression of phospholysine phosphohistidine inorganic pyrophosphate phosphatase [53] (*Lhpp*), Solute Carrier Family 13 Member 3 or Na^+^-dependent dicarboxylate co-transporter [54] (*Slc13a3*), and brain permeability regulator [55] plasmalemma vesicle associated protein (*Plvap*) was increased by AAC2 but reduced by AAC2-hINS. Free hINS mediated similar, albeit modest, downregulation. All these genes regulated brain function and were implicated in neurodegeneration [53,55]. A pattern of regulation for genes, such as *Clec2d* and *Angpl8*, which contrasted with that of the free AAC2 and hINS supports function of AAC2-hINS as a complex because it did not mimic the effects of free AAC2 or hINS.

Distinct pathways were also discovered by a comparison of free hINS and AAC2-hINS treated groups. Treatment with AAC2-hINS induced a key transporter for taurine *Slc6a6* [56,57] and an enzyme carbonic anhydrase *CA14*, but decreased mevalonate kinase (*Mvk*) and glutathione peroxidase *Gpx6* pathways compared to free hINS (Figure 6b). Free hINS reduced *Slc6a6* (Figure 6c, −32% vs. Control 100%) and *CA14* compared to control STZ-induced mice (Figure 6d, −30% vs. Control 100%). In contrast, AAC2-hINS treatment increased expression of these genes, while free AAC2 had no significant effect. A deficiency in *Slc6a6* is associated with chronic liver disease [58], and overall reduction in lifespan [59]. *CA14* is a transmembrane enzyme [60]. It maintains extracellular pH, and facilitates anion transport required for synaptic transmission [61]. The effect of hINS treatment manifested by a marked increase in *Gpx6* expression (290%) compared to all other treatment groups (Figure 6e). Binding of hINS to AAC2 in complex decreased *Gpx6* expression in a similar fashion as free AAC2. *Gpx6* is one of the least understood genes in the family antioxidative glutathione peroxidase enzymes [62] and the increased *Gpx6* expression was found in mice models of Parkinson disease [63] and beta cell models of gestational diabetes [64], whereas its role in the liver has not been established.

AAC2-hINS mediated the synergistic downregulation of *Mvk* (−46% vs. Control) compared to a modest −5% and −8% decrease observed in response to free hINS and AAC2 treatments (Figure 6f). *Mvk* is a regulatory enzyme in the cholesterol synthesis pathway, which increases the risk of coronary artery disease [65] and periodic fever syndromes [66]. The distinctive changes in 60 specific metabolic and regulatory genes in the liver that were observed for the AAC2-hINS complex, but not for either component alone, implied that the complex was sufficiently stable to function uniquely under the in vivo conditions. The observation that the complex induced highly distinct signaling pathways, termed interactome, indicated that the therapeutic impact of AAC2-hINS could not be merely attributed to a scaffolding role for AAC2. Various scaffolds have been currently used for the delivery, protection, and regulation of bioactive proteins, and the creation of macromolecular microenvironments supporting enhanced enzymatic activity [67]. Our findings support a novel mode of action in which AAC2-hINS functions as a complex generating a new interactome. AAC2-hINS complex resembled the action of natural protein complexes which define biological systems by specific interactomes [68].

Although the mechanism underlying AAC2-hINS action involves hepatic genes, the central regulation of metabolism and inflammation in the liver may influence the central nervous system (CNS). In particular, taurin metabolism has been identified as a pathway influencing both chronic liver disease [58] and CNS disorders associated with severe nephropathy in STZ mice [69], retinal degeneration [56], and other neurological dysfunctions [70]. Therefore, we continued with the locomotor and behavioral assessment of the cumulative functional impact of free and bound AAC2 and hINS on CNS function in STZ mice.

### 3.7. AAC2-hINS Improved Cognitive Performance in T1D Mouse Models

Locomotor activity was significantly improved in AAC2-hINS-treated STZ (+60%) mice during the active dark periods, whereas mice treated with free AAC2 reduced their activity (−23%) and free hINS had no effect compared to the control (Figure 7a). We used the Y Maze Spontaneous Alternation Test to assess cognitive function in the STZ mice after 20–25 days of treatment (Figure 7b). A higher alternation percentage indicates better spatial memory function [37]. A significant improvement in spontaneous alteration (% SA) was observed in AAC2-hINS-treated STZ mice. Changes in the groups treated with either free hINS or AAC2 were not significant compared to control groups, even though immunostaining demonstrated the presence of hINS in the brain, e.g., hippocampus, in groups treated with either free hINS or AAC2-hINS complex (Figure 7c). Detection of intact AAC2-hINS complex in the brain was technically unachievable; however, the presence of hINS in the brain and the synergistic locomotor and cognitive performance observations support a specific action of AAC2-hINS treatment and its efficacy in the CNS of STZ-induced T1D mice. 

Given that STZ can alter sensory perception [71], which can influence the results of the Y Maze test, we validated the locomotor and cognitive efficacy of AAC2-hINS vs. free AAC2 and hINS in a genetic *Ins2^Akita^* mouse model of T1D (Figure 7d–g).

In our previous study, we reported the outcomes of treatment with either free AAC2 or hINS on glucose metabolism and showed only a marginal effect on the overall cognitive performance of *Ins2^Akita^* mice after a short (7 weeks) treatment period [31]. In contrast, mice treated with AAC2-hINS in the same study showed a marked increase in locomotor activity compared to the non-treated control, in contrast to the reduced activity in the AAC2-treated mice and a lack of the effect in mice receiving free hINS injections (Figure 7d). The cognitive performance was accessed using a Barnes maze test (Figure 7e–g). The treatment with AAC2-hINS led to a significant and progressive decrease in latency, e.g., time to reach the right escape hole compared to other groups (Figure 7e). The number of errors was significantly different only in AAC2-hINS-treated mice compared to control (Figure 7f). Synergistic cognitive improvement was manifested as 130% longer time spent in the escape hole quadrant (% in path Q3) by mice treated with AAC2-hINS compared to the control *Ins2^Akita^* mouse in the probe trial on day 6 (Figure 7g). Thus, the combination of AAC2 and hINS into a bound complex significantly improved their effects on the CNS that was manifested as improved locomotor and cognitive performance. Complex physiological effects, including activity and cognitive behavior, replicated the same phenomena: they exclusively manifested in mice treated with AAC2-hINS complex, but not the isolated components.

Although deciphering the precise mechanism underlying the action of AAC2-hINS in neural pathways was beyond the goal of this study, IPA analysis identified taurine metabolism dependent on the taurine transporter *Slc6a6* as the major pathway regulated by AAC2-hINS in the liver. Previous research has demonstrated the role of *Slc6a6* in diabetic complications in both humans and rodents [56,59,69]. Moreover, the taurine deficiency underlined the pathogenesis of diabetes [72], particularly those related to cognitive impairments [73]. Taurine directly influenced cholesterol [74] and glucose metabolism [75] as well as insulin secretion and signaling [76,77]. It is plausible that the improved activity, cognition, and the suppression of *Mvk* regulating cholesterol synthesis could be attributed to the expression of *Slc6a6*. Taurine responses depend on the leptin/LepR pathway [75,78] and were disrupted in *LepR^db^* mice and *Lep^ob^* mice [79]. LepR also plays a key role in the regulation of BBB permeability [80] and coordinates the work of other receptors, including InsR [27]. In our previous work, we established that AAC2 functioned as an alternative activator of LepR [31]. In line with this mechanism, AAC2 increased the expression of *Slc6a6* compared to hINS (Figure 6c), but only the AAC2-hINS complex leads to a significant increase in the expression of this transporter and markedly improved cognition and activity in mouse models of T1D. Although the signaling cascade remains to be elucidated, restorative pathways that repair systemic diabetic malfunction and induce neurotransmitter activity likely contributed to the neuroprotective effects of the AAC2-hINS nanofibers.

## 4. Conclusions

The clinical features of different forms of diabetes increasingly overlap [81], especially with regard to neurological degeneration and dementia [82,83,84]. The occurrence of these conditions are high in patients with T1D and T2D [82,83,84] and commonly co-exist with obesity, cardiovascular disease, and inflammation [85,86]. The complex pathogenesis of diabetes requires combinatorial solutions for treatment. In this study, we showed that the noncovalent complex formed between self-assembled peptide nanofibers and natural proteins produced emergent properties that enhanced the therapeutic outcomes in chronic, multifactorial metabolic diseases. We demonstrated an ability of AAC2 nanofibers to bind insulin on the surface in vitro. In vivo, under conditions resembling T1D, the mice treated with the AAC2-hINS complex retained hINS in blood longer without inducing hypoglycemic episodes. The most important advantages of the AAC2-hINS complex were improved glucose uptake in BBB endothelial cells and increased expression of the neurotransmitter taurine transporter, insulin, and other genes in the brain. Cumulatively, these advantages contributed to a higher overall activity and cognition of the mice chemically induced with STZ and genetic T1D treated with the AAC2-hINS formulation, in contrast to AAC2 or hINS alone, which did not exhibit these effects. Our findings revealed that binding insulin with these nanofibers offered a feasible strategy to address the rising prevalence of cognitive diseases in both insulin-deficient T1D and, possibly, insulin-resistant T2D patients.

The ability of the AAC2-hINS complex to achieve a balance between the catabolic responses induced by the regulation of LepR/GLUT1 in neural and immune systems by AAC2 [31] with the anabolic effects of hINS/InsR/GLUT4 axes in peripheral tissues [87], is a critical aspect of this therapeutic regimen. In our studies, treatment with AAC2-hINS produced physiological outcomes that approached the parameters seen in healthy mice (Appendix A). Refinement of the dose, regimen, and stability/design of the complex could further improve this energy balance and the corresponding efficacy of the treatment. The cross-regulation of LepR and InsR for catabolic and anabolic glucose utilization is known as the combinatorial principle in the regulation of glucose metabolism [21,27] and contrasts with insulinocentric therapies for T1D [1]. Arguably, deficiencies in neurodevelopment in children [16,17] and neurodegeneration in patients with diabetes are one of the most complex and harmful complications of diabetes [83,88,89]. Our data support that AAC2 could serve as a new small molecule prototype for binding insulin and for the combinatorial treatment of diabetes and its complications.

The modular structure of AAC2 nanofibers permits effective binding of insulin and other potential biologics. The flexible design of the AAC class of molecules could be used for developing a variety of combination therapeutics for personalized treatment of diabetes and its complications.

## Figures and Tables

**Figure 1 pharmaceutics-14-00081-f001:**
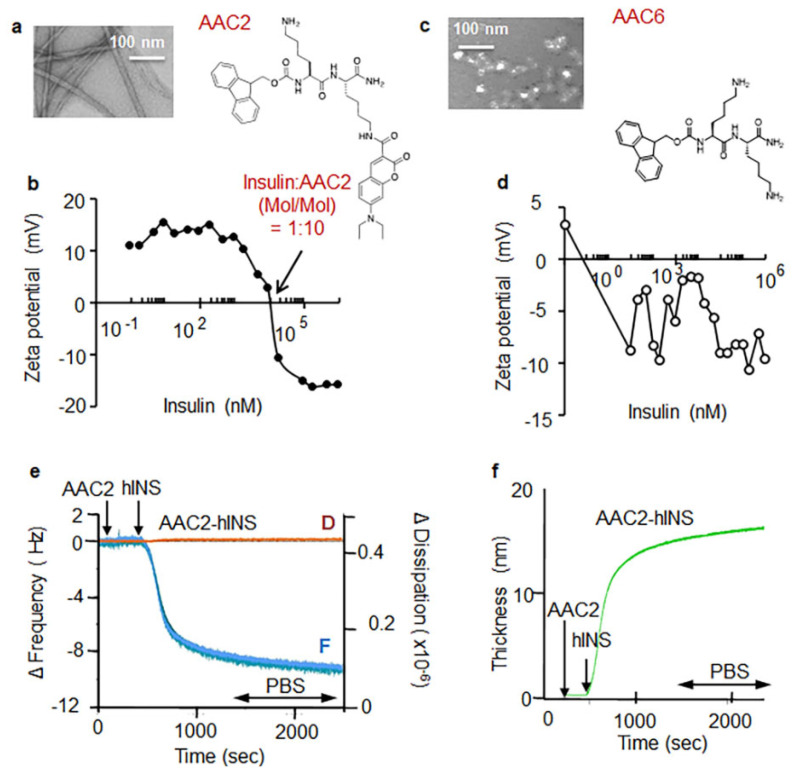
AAC2 is self-assembled into a positively charged nanofiber, which electrostatically bound insulin. (**a**,**c**) The morphology (TEM at 225,000 magnification) and chemical structure of AAC2 (**b**,**d**) Electrostatic interaction between AAC2 (0.1 mM in PBS) (**b**) or AAC6 (**d**) and negatively charged hINS measured as changes in Zeta potential (mV). 1:10 hINS:AAC2 (mol/mol) indicated 0 mV. (**e**) Binding kinetics of AAC2 (0.1 µM in PBS) and hINS (10 µg/mL in PBS) onto the active gold sensor was measured as changes in frequency (F, blue line) and dissipation (D, orange line) of the acoustic wave using quartz crystal microbalance (QCM). AAC2-hINS film was stable during PBS wash (arrow). (**f**) The thickness of AAC2-hINS film (15 nm) was calculated based on F and D kinetics.

**Figure 2 pharmaceutics-14-00081-f002:**
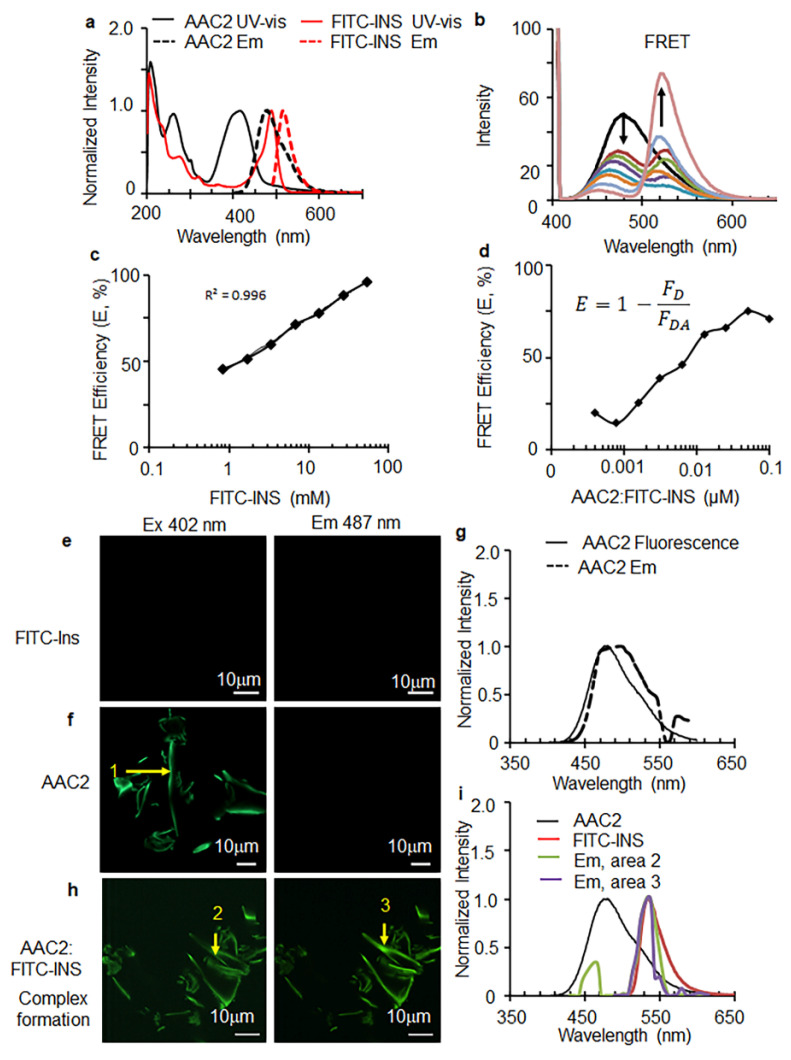
Fluorescence resonance energy transfer (FRET) between interacting AAC2 and Fluorescein (FITC)-labeled INS (FITC-INS). (**a**) Normalized UV-vis absorbance and fluorescence excited (Ex) at 402 nm intensity of AAC2 (1 mM) and FITC-INS (57.3 μM for UV-vis and 28.65 μM for fluorescence Ex 487 nm) in PBS to show the spectral overlap. (**b**) Fluorescence spectra of the 1:0.67 AAC2:FITC-labeled insulin mixtures were diluted serially in PBS. (**c**) The percentage of FRET efficiency of each mixture was calculated based on (**b**) using the emission at 479 nm of the mixtures and pure AAC2. (**d**) The percentage of FRET efficiency was calculated based on the emission at 479 nm of fluorescence spectra (Ex 405 nm) of mixtures of 0.1 mM AAC2 with varying concentrations of FITC-labeled insulin to obtain different molar ratios, with a constant concentration of AAC2. Mixtures were incubated at room temperature for 30 min prior to measurement. (**e**,**f**,**h**) Confocal images and emission spectra of (**g**,**i**) of 0.1 mM FITC-INS (**e**), 1 mM AAC2, (**f**,**g**) and (**h**,**i**) 10:1 AAC2:FITC-INS (containing 1 mM AAC2) was incubated at room temperature for 30 min prior to imaging. Each sample was excited at both 402 nm (first column) and 487 nm (second column). The emission data were co-plotted with normalized emission, comparing AAC2 and FITC-INS. The dashed line indicates the emission of area 1 in the confocal image.

**Figure 3 pharmaceutics-14-00081-f003:**
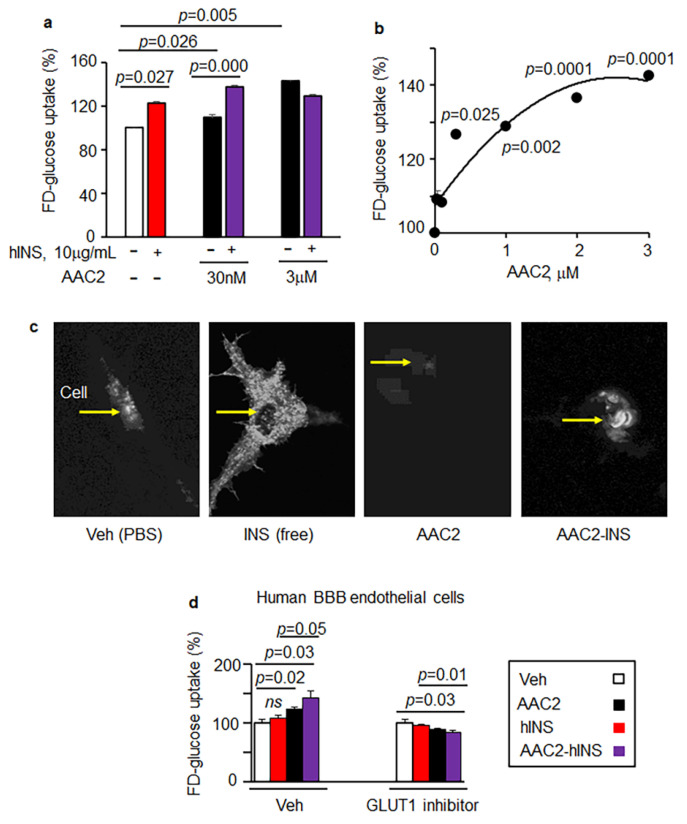
AAC2-hINS increases glucose uptake in 3T3-L1 preadipocytes and human BBB endothelial cells via GLUT4 and GLUT1. (**a**) FD-glucose uptake in mouse 3T3-L1 preadipocytes (*n* = 24 per treatment) treated with vehicle (Veh; PBS, open bar) or hINS (1.7 µM, red bar) or AAC2 (0.03 and 3 µM black bar) or AAC2-hINS (purple bar) with the respective concentration of AAC2 and a constant hINS concentration. After 2 h 15 min starvation, FD glucose and treatment reagents were added for an additional 85 min of incubation. Data (%) were normalized to FD-glucose uptake in control (Veh) 3T3-L1 cells (100%). Student’s independent *t*-test for all tests in figure. (**b**) AAC2 dose-dependent uptake FD-glucose in 3T3-L1 preadipocytes. Data are shown as % of increase compared to non-stimulated cells. (**c**) Non-differentiated NIH-3T3 cells were transiently transfected with GLUT4-GFP. Translocation of GLUT4-GFP (white, yellow arrow) cytosol to the membrane after stimulation with Veh (PBS), hINS (10 µg/mL), AAC2 (10 µM), or AAC2-hINS was demonstrated using confocal microscopy (60× magnification). (**d**) FD-glucose uptake in human brain endothelial cells (*n* = 7 per treatment) treated with vehicle (Veh; PBS, open bar) or hINS (1.7 µM, red bar) or AAC2 (0.1 µM black bar), or AAC2-hINS (purple bar) in the presence and absence of GLUT1 inhibitor (BAY-876; 10 nM in DMSO). Cells were pre-treated with or without BAY-876 for 40 min and then FD glucose and treatment reagents were added for an additional 50 min of incubation. Data (%) were normalized to FD-glucose uptake in control (Veh without inhibitor, 100%).

**Figure 4 pharmaceutics-14-00081-f004:**
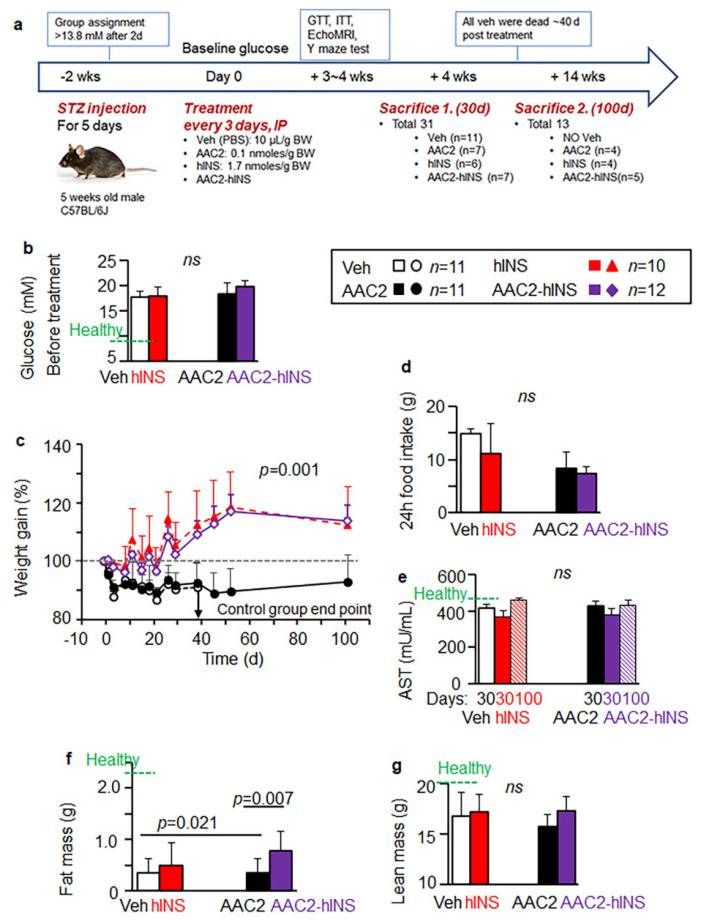
Improved body composition in AAC2-hINS-treated mice with STZ-induced T1D. (**a**) Schematic presentation of induction of experimental T1D by streptozotocin (STZ mice, total *n* = 44 mice) and treatment protocol of control group with vehicle, AAC2, hINS, AAC2-hINS complex. Treatments were administered intraperitoneally (i.p.) every 3 days. After starting treatments, mice died or were sacrificed prior to the death (30 days group). The surviving mice continued with treatments for 100 days. (**b**) Glucose levels (mM) in STZ-induced mice groups prior to the treatment. One-way ANOVA. *p* = 0.157; not significant, ns. Green dashed line-the fasting glucose levels in healthy mice. (**c**) Weight gain of the STZ mice was measured until they succumbed to the diseased (*n* = 7) or were sacrificed. Data are shown as % of baseline body weight of each animal before induction of T1D (100%, gray dash line). Overall comparison *p* = 0.001 (Kruskal–Wallis test with a Mann–Whitney U-test for post hoc testing). For (**d**) Food intake was collected for 24 h on multiple days. (**e**) Plasma aspartate aminotransferase (AST) levels in plasma of the STZ mice were measured after 30 days and 100 days post treatment. One-way ANOVA. ns, not significant. Green dashed line-the AST levels in healthy mice (*n* = 6). (**f**,**g**) Fat (**f**) and lean (**g**) body mass of the STZ mice and healthy mice (dashed line, *n* = 6) were measured using Echo-MRI 3 weeks after. *p* = 0.043 for overall comparison of fat body mass. ns, ANOVA.

**Figure 5 pharmaceutics-14-00081-f005:**
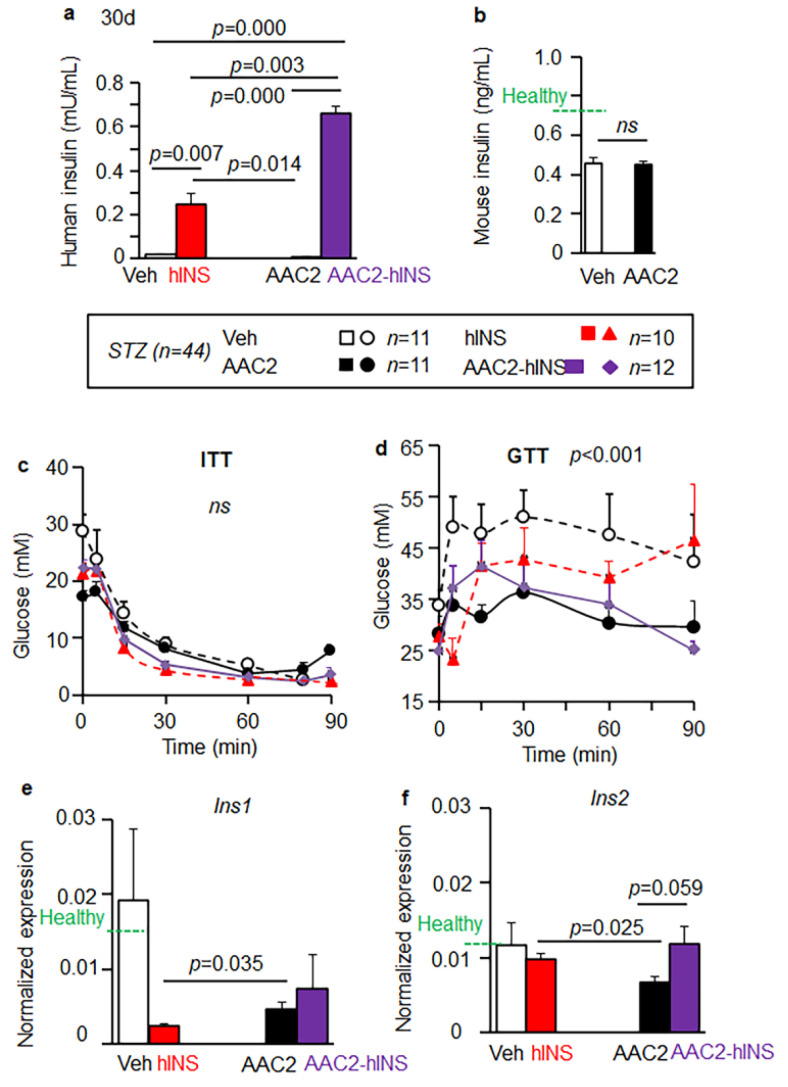
AAC2-hINS treatment vs. free hINS increased levels of circulating therapeutic hINS and expression of endogenous insulin in the brain. (**a**) The plasma levels of therapeutic hINS were measured three days after the last treatment (30 d). For overall comparison, a Kruskal–Wallis test with a Mann–Whitney U-test for post hoc testing, *p* = 0.001. (**b**) The plasma levels of endogenous mouse insulin mice were measured in the same samples. n.s.—Student’s independent *t*-test. (**c**) Insulin tolerance test (ITT) in the STZ mice was performed one day after treatment administration. One-way ANOVA. ns. (**d**) Glucose tolerance test (GTT) in STZ mice. Overall comparison, *p* = 0.001. (**e**,**f**) Expression levels of mouse *Ins1* (**e**) or *Ins2* (**f**) in the STZ mouse brain tissues were measured by a Taqman Mann–Whitney U test. Dashed line, levels in healthy mice (*n* = 6).

**Figure 6 pharmaceutics-14-00081-f006:**
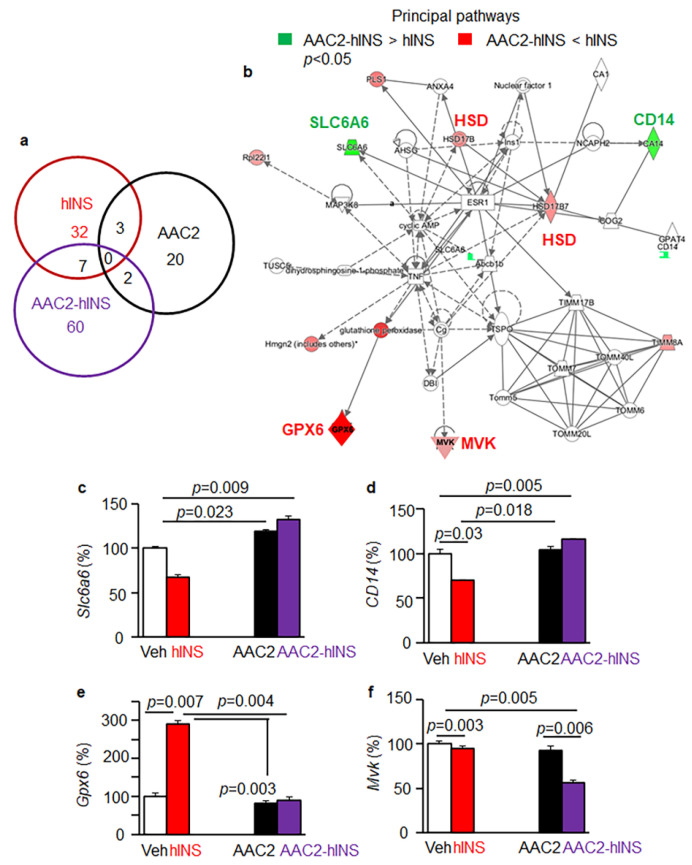
AAC2-hINS treatment regulated a distinct set of genes compared to its free constituents. (**a**) Comparison of unique and overlapping the expression of hepatic genes, that were significantly different from the control (Veh, not shown, *n* = 3, cut off were ≥1.5 fold difference and *p* < 0.01) across STZ mice treated with AAC2 (black), hINS (red), and AAC2-hINS (purple), (*n* = 3/group). Gene expression was analyzed by Affymetrix GeneChip. (**b**) Ingenuity pathway analysis was performed based on the statistically different genes predicted metabolic hubs that were distinctly regulated in response to hINS vs. AAC2-hINS treatments. Green shapes indicate higher expression in AAC2-hINS vs. hINS groups, such as *Slc6a6* (**c**) and *CD14* (**d**). Red shapes show the higher expression of genes in IPA analysis in mice treated with free hINS vs. AAC2-hINS, such as *Gpx6* (**e**) and *Mvk* (**f**).

**Figure 7 pharmaceutics-14-00081-f007:**
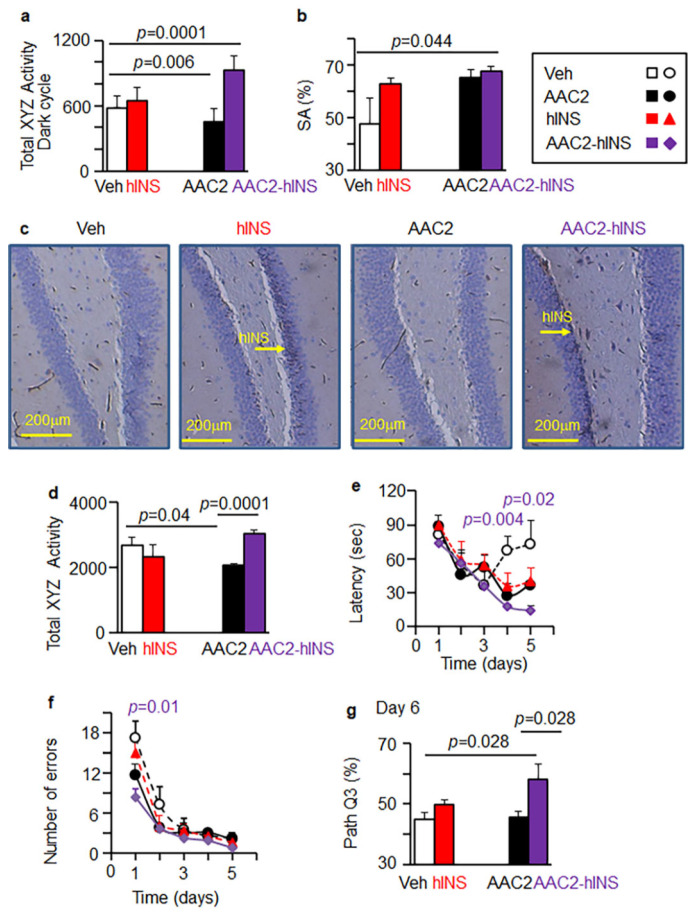
AAC2-hINS treatment improved physical activity and cognitive performance in STZ and genetic mouse models of T1D. (**a**) Activity of mice in XYZ directions in STZ mice was measured during the active dark period using CLAMS. Student’s *t*-test. (**b**) Spontaneous alteration (% SA) in STZ mice was measured 3–4 weeks post treatment. One-way ANOVA. (**c**) Representative images of protein expression of hINS (brown staining, yellow arrows) in the hippocampus of STZ mice treated with Veh, hINS, AAC2, AAC2-hINS (*n* = 3/group). (**d**) Activity of mice in XYZ directions in *Ins2^Akita^* mice (*n* = 5/group) was measured during the active dark period using CLAMS. Student’s *t*-test. (**e**,**f**) Barnes Maze test was performed with *Ins2^Akita^* mice after 7 weeks post-treatment (*n* = 5/group). In the training period (Day 1~5), latency (**e**) and a number of errors (**f**) were measured. Differences were compared for each day by Student’s independent *t*-test. Significant differences between control and AAC2-hINS-treated mice on days 4 and 5 were *p* = 0.004 and 0.02, respectively. At baseline, the control group presented significantly more errors compared to the AAC2-hINS treated group (*p* = 0.01). (**g**) Hole escape time at day 6 (Q3 in same Barnes Maze experiment). Overall comparison *p* = 0.026.

## Data Availability

Data will be available on request according to “MDPI Research Data Policies” at https://www.mdpi.com/ethics, accessed on 18 October 2021.

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
