# Peer review of "Amino Acid Nanofibers Improve Glycemia and Confer Cognitive Therapeutic Efficacy to Bound Insulin"

_pharmaceutics, 2021, doi:10.3390/pharmaceutics14010081_

Round 1
Reviewer 1 Report
This paper reports the development of positively charged amino acid-modified nanofibers to bind insulin to improve glycemia. AAC2 amino acids could self-assemble into nanofibers with a positively charged surface. The nanofibers could bind to negatively charged insulin and induce combinatorial effects on the regulation of glucose metabolism in both nervous and peripheral tissue, causing mitigating a decline in mice with insulin-deficient diabetes. The formation of self-assembled nanofibers was confirmed by TEM analysis, while Zeta potential, FRET, and QCM-D analyses confirmed the binding of insulin to nanofibers. In vitro studies showed that the binding of free insulin to the nanofibers could boost the blood levels of insulin and glucose uptake in mice.
I suggest the acceptance of the manuscript after the below minor issues are addressed.
- Line 112, use the full name for PBS (phosphate-buffered saline) when it is used for the first time.
- Ref 18, add the missing issue and page numbers (97 (3) e275-e283; DOI: 10.1212/WNL.0000000000012243)
- Ref 32, correct the DOI numbers (https://doi.org/10.1238/Physica.Regular.059a00391)
- Refs 35 and 44, DOI numbers are missing
- Ref 90, add missing issue and page numbers (Nature Reviews Neurology volume 14, pages168–181 (2018)
Author Response
Reviewer 1.Comments and Suggestions for Authors
This paper reports the development of positively charged amino acid-modified nanofibers to bind insulin to improve glycemia. AAC2 amino acids could self-assemble into nanofibers with a positively charged surface. The nanofibers could bind to negatively charged insulin and induce combinatorial effects on the regulation of glucose metabolism in both nervous and peripheral tissue, causing mitigating a decline in mice with insulin-deficient diabetes. The formation of self-assembled nanofibers was confirmed by TEM analysis, while Zeta potential, FRET, and QCM-D analyses confirmed the binding of insulin to nanofibers. In vitro studies showed that the binding of free insulin to the nanofibers could boost the blood levels of insulin and glucose uptake in mice.I suggest the acceptance of the manuscript after the below minor issues are addressed.
We would like to thank this reviewer for his/her detailed revision.
- Line 112, use the full name for PBS (phosphate-buffered saline) when it is used for the first time.
We added the full name.
- Ref 18, add the missing issue and page numbers (97 (3) e275-e283; DOI: 10.1212/WNL.0000000000012243)
We added missing issue and page numbers
- Ref 32, correct the DOI numbers (https://doi.org/10.1238/Physica.Regular.059a00391)
We corrected DOI
- Refs 35 and 44, DOI numbers are missing
DOI was added to Ref 35, but ref 44 has no DOI
- Ref 90, add missing issue and page numbers (Nature Reviews Neurology volume 14, pages168–181 (2018)
We updated this information.
Reviewer 2 Report
In this work Lee A and colleagues evaluated the therapeutic effect of AAC2-hINS in diabetic mice. First, they have characterized the complex regarding its physico-chemical properties and in vitro performance. Next, the authors have demonstrated the advantage of using such fibrillar complex in controlling STZ-induced diabetes, including on 1) increased levels of circulating INS, 2) and a “normalized” GTT profile. The authors further evaluated the biochemical changes on mice and correlated such with behavior test results. Overall, the results support a positive effect of AAC2-hINS on controlling diabetes with a possible positive neurological effect. Despite the very interesting and promising results, I believe the manuscript requires further revision before publication. I recommend the manuscript to be returned to the authors with major revisions. Please consider the following:
OVERALL
1 – Overall, I found the results and discussion sections to be unbalanced. The results section, complete in its whole, was very extensive and detailed. However, such level of detail might lead the readers to lose the context of the work. I recommend improving (and simplify if possible) this section by 1) adding context to the experiments performed and molecular targets/indicators used, 2) highlight the main findings from each set of experiments and 3) provide clues to the next logical steps to take.
2 – Also consider merging the results and discussion section for a better understanding of the work.
3 – The captions used for the panels are extremely rich in information but make them very hard to read and understand. Please consider reducing this.
MATERIALS AND METHODS
4 – In phrase 200-201, insert the reference values for normoglycemia in mice.
5 – Regarding mRNA analysis, can you explain your choice of house-keeping gene?
6 – Regarding the values presented in phrase 400-401, can you improve their presentation for a better understanding?
7 – Regarding the cell assays, your claim to testing the complex with “brain cells” (phrase 413) is restricted to endothelial cells. It is true these cell play an important role in brain metabolism. However, it is also true that insulin can cross the BBB and influence both neurons and astrocytes. Please consider changing this reference to “brain cells” for another more appropriate one.
8 – In figure 3d, the results seem to indicate no differences between untreated and BAY-876-treated cells. Can you comment on this?
9 – Regarding the caption of figure 4, phrase 449, please reconsider changing the phrase “ mice were died…”.
10 – Regarding phrase 477, your results show that AAC2 treated mice lost weight and its recovery seem to be really slow, contrasting with hINS and AAC2-hINS. Please, consider rephrasing these ideas to match your results.
11 – I did not find the supplementary videos referenced in the manuscript. Can you provide them again?
12 – Regarding section 3.4., how much of AAC2-hINS induced increased insulin is induced by the complex and not AAC2-hINS dissociation? Was the dose of insulin provided as hINS and AAC2-hINS the same? Please, comment on this and insert this information.
13 – Regarding section 3.4., consider explaining the importance of Ins1/2 gene for your work. Also consider doing this for all the targets analyzed.
14 – Regarding the gene expression results for the mice, you analyze hepatic tissue but not brain tissue. Yet, you provide a possible mechanism of neuroprotection for the complex. Can you provide a similar analysis to brain tissue samples of mice? If not, please consider changing this section to highlight that these results concern the liver alone.
15 – What was the long-term effect of AAC2, hINS and AAC2-hINS on glycated hemoglobin? I believe this indicator is important to evaluate the effectiveness of your complex in maintaining normoglycemia.
Author Response
Answers to the reviewer 2
Comments and Suggestions for Authors
In this work Lee A and colleagues evaluated the therapeutic effect of AAC2-hINS in diabetic mice. First, they have characterized the complex regarding its physico-chemical properties and in vitro performance. Next, the authors have demonstrated the advantage of using such fibrillar complex in controlling STZ-induced diabetes, including on 1) increased levels of circulating INS, 2) and a “normalized” GTT profile. The authors further evaluated the biochemical changes on mice and correlated such with behavior test results. Overall, the results support a positive effect of AAC2-hINS on controlling diabetes with a possible positive neurological effect. Despite the very interesting and promising results, I believe the manuscript requires further revision before publication. I recommend the manuscript to be returned to the authors with major revisions. Please consider the following:
We would like to thank this Reviewer for his/her detailed revision.
OVERALL
1 – Overall, I found the results and discussion sections to be unbalanced. The results section, complete in its whole, was very extensive and detailed. However, such level of detail might lead the readers to lose the context of the work. I recommend improving (and simplify if possible) this section by 1) adding context to the experiments performed and molecular targets/indicators used, 2) highlight the main findings from each set of experiments and 3) provide clues to the next logical steps to take.
2 – Also consider merging the results and discussion section for a better understanding of the work.
Based on the reviewer’s suggestions we merged Results and Discussion into one section.
3 – The captions used for the panels are extremely rich in information but make them very hard to read and understand. Please consider reducing this.
We reduced the wording, unnecessary details of the measurements, and group description stated in the legends.
MATERIALS AND METHODS
4 – In phrase 200-201, insert the reference values for normoglycemia in mice.
We inserted the reference value from healthy mice.
5 – Regarding mRNA analysis, can you explain your choice of house-keeping gene?
HPRT was used as housekeeping gene in the liver. We validated its stable expression compared with 18S, Gapdh, and tata box binding protein (Tbp) gene.
6 – Regarding the values presented in phrase 400-401, can you improve their presentation for a better understanding?
We rephrase this paragraph: ‘FD-glucose uptake reached a plateau at higher concentrations of AAC2 with or without hINS; however, AAC2 binding with hINS significantly increased uptake of FD-glucose at nanomolar concentrations of free AAC2 (Figure 3a). Free AAC2 exhibited a dose-dependent uptake of glucose (Figure 3b)’.
7 – Regarding the cell assays, your claim to testing the complex with “brain cells” (phrase 413) is restricted to endothelial cells. It is true these cell play an important role in brain metabolism. However, it is also true that insulin can cross the BBB and influence both neurons and astrocytes. Please consider changing this reference to “brain cells” for another more appropriate one.
We changed this sentence to: ’We also analyzed the effect of free AAC2, hINS and AAC2-hINS in brain endothelial cells (bEC), which utilize GLUT1 transporter that is not dependent on insulin [44]’.
8 – In figure 3d, the results seem to indicate no differences between untreated and BAY-876-treated cells. Can you comment on this?
BAY-876 has an intrinsic fluorescence Siebeneicher et al., Identification and Optimization of the First Highly Selective GLUT1 Inhibitor BAY‐876. ChemMedChem. 2016 PMID: 27552707. The structure is provided below. BAY-876 also influences the basic cellular metabolism. Both factors result in the increase in background fluorescence. However, all leptin, AAC2, and AAC2-hINS stimulation led to the decreased FD-glucose uptake compared to control cells in the presence of BAY-876. In the revised Figure 3d, we normalized background absorption (100%) in all samples containing BAY-876, to avoid this lengthy explanation, in line with this Reviewer’s suggestion.
9 – Regarding the caption of figure 4, phrase 449, please reconsider changing the phrase “ mice were died…”.
We replace this word with ‘succumbed to the disease’.
10 – Regarding phrase 477, your results show that AAC2 treated mice lost weight and its recovery seem to be slow, contrasting with hINS and AAC2-hINS. Please, consider rephrasing these ideas to match your results.
Thank you for pointing that out. We rephrase this statement to:’ Mice treated with free AAC2 gained significantly less weight compared to the other treatment groups receiving free and bound hINS; however, they remain functional throughout the 100-day intervention’.
11 – I did not find the supplementary videos referenced in the manuscript. Can you provide them again?
I will resubmit videos and work with the journal to ensure their availability.
12 – Regarding section 3.4., how much of AAC2-hINS induced increased insulin is induced by the complex and not AAC2-hINS dissociation? Was the dose of insulin provided as hINS and AAC2-hINS the same? Please, comment on this and insert this information.
We provided this information in Methods. We included a rationale for the selections of these concentrations that was based on the experiments in cells rather than of FRET studies showing dissociation of AAC2-hINS. The dissociation studies used high concentrations of these molecules, which were incompatible with therapeutic doses. We also added the discussion stating analytical techniques quantified the presence of native protein complexes in the physiological environments; however, biological evidence supported a specific function of AAC2-hINS compared to its free cosntituents. Both statements are provided below for your convinience.
Given that the imroved glucose uptake efficacy in 3T3-L1 cells has been seen at nanomolar concentrations of AAC2 (30nM) and milimolar concentrations of insulin (1.7mM) (Fig. 3a), we used 1:17 AAC2: hINS mol/mol ratio in AAC2-hINS and used the same dose for free AAC2 and hINS in this study.
It is possible that the concentrations used in our studies were suboptimal and AAC2-hINS complex could be stabilized with the increased molar ratio of AAC2 based on the Zero potential and FRET studies and could be improved in the future. The dose of AAC2:hINS (1:17, mol/mol) used in this study could lead to dissociation of complex. However, in our study, the presence of scaffold was strongly supported by the markedly different stabilities of the bound and free forms of hINS in the circulation. AAC2-hINS formulation circulated in blood at 2.7 higher concentrations than free hINS 48h after the injection. The presence of the AAC2-hINS complex partially account for these differences in concentrations over time as well as the absence of hypoglycemic episodes, a known side effect of free insulin overdose [1]. Establishing the stability and the tissue distribution of the native AAC2-hINS complex under the low therapeutic concentrations used in this study remains technically challenging. Currently, some predicted complexes were re-solved in artificial ‘native’-like environments using cryogenic electron microscopy re-quired a specific milieu [70], whereas isolation and quantification of these native com-plexes from tissues has not been successful. In our subsequent experiments, we compered gene expression to elucidate AAC2:hINS complex regulate similar pathways as its free AAC2 and hINS constituents or act a distinct therapeutic modality.
13 – Regarding section 3.4., consider explaining the importance of Ins1/2 gene for your work. Also consider doing this for all the targets analyzed.
We added the importance of this work to this paragraph and a summary sentence.
14 – Regarding the gene expression results for the mice, you analyze hepatic tissue but not brain tissue. Yet, you provide a possible mechanism of neuroprotection for the complex. Can you provide a similar analysis to brain tissue samples of mice? If not, please consider changing this section to highlight that these results concern the liver alone.
The brain tissues are very heterogenous and the expression studies should be performed in specific section of the brain. This will required a focused study. Liver is a metabolic hub responding, to metabolic, toxic, neural, and inflammatory stimuli from all tissues, including brain. We removed all descriptions related to CNS responses. We also conclude this section with the modified sentence:‘Although the mechanism underlying AAC2-hINS action involves hepatic genes, the central regulation of metabolism and inflammation in the liver may influence with established functions in the central nervous system (CNS)’.
15 – What was the long-term effect of AAC2, hINS and AAC2-hINS on glycated hemoglobin? I believe this indicator is important to evaluate the effectiveness of your complex in maintaining normoglycemia.
HbA1C is a marker of glucose changes for 3 months, therefore, the 30 days of treatment yield significant
results in mice treated with AAC2-hINS (Mix yellow circle, Figure below, P<0.03 compared to control); however, the other data were inconclusive. The Figure for this Reviewer shows the HbA1C levels 30 days and 100 days post-treatment in all groups. 30 days after the treatment the levels of HbA1C were significantly different only between control and AAC2-hINS (Mix yellow circle)
In the course of remaining days, the highest significant increase (P<0.05, paired t-test) was observed only in mice treated with free hINS (gray line, gray circle), while mice treated with free AAC2, or AAC2-hINS have a similar moderate increase. The mice in control STZ group did not survive. Therefore, these data remain inconclusive, and we request not to include them in the manuscript.

Round 2
Reviewer 2 Report
In this first set of revisions Lee A and colleagues addressed most of my concerns and improved the manuscript. However, there were some key-points that require more attention. If the address these, I believe the manuscript can be published. Consider the following:
1 – Regarding point 6 of my last revision, I want to clarify that I was addressing the phrase “hINS 123%, AAC2 109%-143%, AAC2-hINS 137% vs. 100% control; Figure 3a” (now phrase 397-398). Please improve this as their interpretation is not easy nor evident.
2 – Regarding point 14 of my last review, I believe you can expand this section as long as it is an evident part of the discussion (e.g. other research avenues) and not result interpretation.
3 – Regarding point 15 of my last revision, I understand the authors position on not showing these results and I am fine with their decision. However, I also advise the authors reconsider and insert this set of results in SI, as it shows another advantage of the AAC2-hINS fibrils over free hINS. Again, I am fine with either option chosen by the authors.
Author Response
In this first set of revisions Lee A and colleagues addressed most of my concerns and improved the manuscript. However, there were some key-points that require more attention. If the address these, I believe the manuscript can be published. Consider the following:
We would like to thank this reviewer for the thoughtful comments, which has improved the clarity of our manuscript. We addressed all the points raised in by the reviewer.
1 – Regarding point 6 of my last revision, I want to clarify that I was addressing the phrase “hINS 123%, AAC2 109%-143%, AAC2-hINS 137% vs. 100% control; Figure 3a” (now phrase 397-398). Please improve this as their interpretation is not easy nor evident.
We misunderstood your previous request. We provided the detailed description in the revised MS.
To test how the formation of the AAC2-hINS complex altered the glycemic properties of AAC2 and hINS, we measured the uptake of FD-glucose by 3T3-L1 mouse fibroblasts/preadipocytes. Compared to non-treated cells (control, 100%), glucose uptake was significantly improved by treatment with free hINS (123%), as well as by free AAC2 at low (30 nM, 109%) and high concentrations (3 mM, 143%) (Figure 3a). Using a constant hINS concentration (10 mg/mL), in conjunction with AAC2, at 30 nM and 3 mM, led to an increase in glucose uptake to 137% and 129 %,respectively. These data suggest that FD-glucose uptake reached a plateau at higher concentrations of AAC2 with or without hINS.Importantly, that AAC2-hINS complex achieve this maxumql effect at low nM concentration of AAC2.
2 – Regarding point 14 of my last review, I believe you can expand this section as long as it is an evident part of the discussion (e.g. other research avenues) and not result interpretation.
We expanded this section by discussion of additional research avenues. Lines 639-642.
Although the mechanism underlying AAC2-hINS action involves hepatic genes, the central regulation of metabolism and inflammation in the liver may influence the central nervous system (CNS). In particular, taurin metabolism has been identified as a pathway influencing both chronic liver disease [58] and CNS disorders associated with severe nephropathy in STZ mice [69], retinal degeneration [56], and other neurological dysfunctions [70]. Therefore, we continued with the locomotor and behavioral assessment of the cumulative functional impact of free and bound AAC2 and hINS on CNS function in STZ mice.
3 – Regarding point 15 of my last revision, I understand the authors position on not showing these results and I am fine with their decision. However, I also advise the authors reconsider and insert this set of results in SI, as it shows another advantage of the AAC2-hINS fibrils over free hINS. Again, I am fine with either option chosen by the authors.
Thank you for your suggestion. Now we provide these data as supplementary materials (new Supplementary Figure 1).